# MVSDet: Multi-View Indoor 3D Object Detection via Efficient Plane Sweeps

**Yating Xu**[1]     **Chen Li**[2,3*]     **Gim Hee Lee**[1]

Department of Computer Science, National University of Singapore[1]
Institute of High Performance Computing, A*STAR[2]
Centre for Frontier AI Research, A*STAR[3]

xu.yating@u.nus.edu   lichen@u.nus.edu   gimhee.lee@comp.nus.edu.sg

## Abstract

The key challenge of multi-view indoor 3D object detection is to infer accurate geometry information from images for precise 3D detection. Previous method relies on NeRF for geometry reasoning. However, the geometry extracted from NeRF is generally inaccurate, which leads to sub-optimal detection performance. In this paper, we propose MVSDet which utilizes plane sweep for geometry-aware 3D object detection. To circumvent the requirement for a large number of depth planes for accurate depth prediction, we design a probabilistic sampling and soft weighting mechanism to decide the placement of pixel features on the 3D volume. We select multiple locations that score top in the probability volume for each pixel and use their probability score to indicate the confidence. We further apply recent pixel-aligned Gaussian Splatting to regularize depth prediction and improve detection performance with little computation overhead. Extensive experiments on ScanNet and ARKitScenes datasets are conducted to show the superiority of our model. Our code is available at `https://github.com/Pixie8888/MVSDet`.

## 1   Introduction

Indoor 3D object detection is a fundamental task in scene understanding and has wide applications in robotics, AR/VR equipment, *etc*. Although point cloud based 3D objection methods [13, 14, 15] have achieved impressive performance, depth sensors are required to capture the data, which may not be available due to budget limitation, form factor constraints, *etc*. Recently, the more economic pipeline of 3D object detection from only posed multi-view images is gaining increasing attention. However, it is much more sophisticated to estimate geometry information from 2D images alone.

A straightforward solution to this problem is using ground truth geometry information, *e.g.* point cloud [19] or TSDF [17], to supervise the model. Built on the 3D volume representation [16], ImGeoNet [19] predicts the emptiness of each voxel by converting the ground truth point clouds to surface voxels as supervision. CN-RMA [17] first reconstructs 3D scenes using ground truth TSDF as supervision and then runs an existing point cloud based object detector to predict bounding boxes. Although they achieve promising performance, the precise ground truth scene geometry is hard to obtain and may not be available [1].

An alternative way is to learn geometry via self-supervision. The pioneer work ImVoxelNet [16] unprojectes 2D image features to a 3D volume representation. However, 2D features can propagate to irrelevant 3D locations since depth information is not known. NeRF-Det [22] relies on a Neural Radiance Field (NeRF) [12] to learn a density field and queries an opacity score for each voxel. The

---

*Chen Li was at the National University of Singapore when this work was done.

38th Conference on Neural Information Processing Systems (NeurIPS 2024).

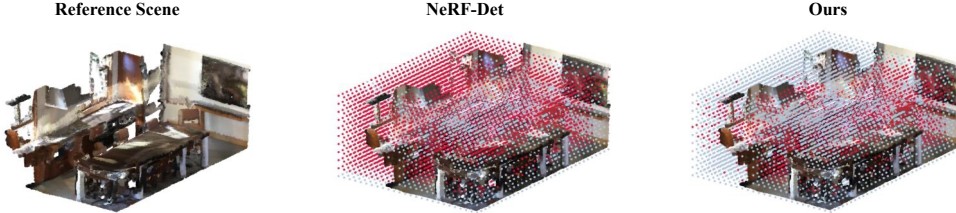

| Reference Scene | NeRF-Det | Ours |

Figure 1: Comparison with NeRF-Det [22]. The 3D voxel centers (grey dots) are overlaid with the reference scene. The red dots denotes the erroneous backprojection pixel features to the points in the free space. Compared to NeRF-Det, we show much less inaccurate backprojections.

opacity score is multiplied with voxel feature to decrease the influence of voxels in the empty space to the feature volume. Since the detection performance is completely determined by the quality of NeRF, enormous effort is spent on making NeRF generalizable and avoiding aliasing issue. Unfortunately, the geometry extracted from NeRF remains unsatisfactory due to insufficient surface constraints in the representation [21]. Consequently, it wrongly backprojects features to voxels in the free space (as illustrated by the red dots in the example shown on the middle of Fig. 1).

In this paper, we propose MVSDet to extract geometry information from only the input multi-view images. A straightforward way is to leverage multi-view stereo algorithms [24, 25] to decide the accurate depths for correct placements of 2D features of each image to the 3D volume. However, accurate depth estimation in the plane-sweeping algorithm [8] requires computationally expensive sampling of many depth planes over all the multi-view images. We mitigate the computational complexity by proposing a probabilistic sampling and soft weighting mechanism to decide the possible depth locations for each pixel. Specifically, we sample multiple top scoring depth proposals in the probability volume that are most likely covering the true depth locations. Pixel features are placed onto the 3D volume only when the backprojected ray intersects at the depth locations. Since the normalized probability score indicates the confidence of the the current depth location, we use it to weigh the pixel feature before assigning the feature to its backprojected voxel center.

To further improve the depth prediction accuracy, we utilize the recent pixel-aligned Gaussian Splatting (PAGS) [3, 28] for novel view rendering as an additional supervision. PAGS predicts a 3D Gaussian primitive [9] for every pixel in the input views, and all the Gaussians are used to render novel views via rasterization-based splatting. Compare to NeRF that uses computation expensive volumetric sampling, Gaussian Splatting is fast and light-weight. A key to good rendering quality in PAGS is the correct positioning of the 3D Gaussians, which depends on an accurate depth prediction. As a result, by putting the Gaussians according to the depth map computed from the probability volume in the plane sweep module, the rendering loss would guide the the Gaussian centers and consequently the depths to the correct values.

In summary, our contributions are as follows:

1. We propose a probabilistic sampling and soft weighting mechanism to efficiently learn geometry without sampling many depth planes in multi-view stereo. Multiple depth proposals are sampled with the probability scores to guide the propagation of image features to 3D voxels.

2. We adopt pixel-aligned Gaussian Splatting to enhance depth prediction without much additional computation overhead, which consequently improves detection performance.

3. We conduct extensive experiments on the ScanNet and ARKitScene datasets to verify the effectivess of our method. Notably, we achieve significant improvements of +3.9 and +5.1 under the mAP@0.5 metric on ScanNet and ARKitScenes, respectively.

## 2  Related Work

**Indoor 3D Object Detection.**    3D object detection for indoor scenes predicts three dimensional bounding boxes and corresponding classes by taking in 3D or 2D inputs. Point cloud is the most popular choice of 3D data for detection as it provide accurate 3D information. To detect objects

from the irregularity and sparseness of the point cloud, VoteNet [14] utilizes Hough voting by points sub-sampling, voting and grouping to generate object proposals. Later methods improve VoteNet by either predicting geometric primitives [27] or using hierarchical graph network [4]. 3DETR [13] reduces hand-coded designs of VoteNet with transformer encoder-decoder blocks. Despite their promising performance, depth sensors are required to capture the data, which is not always available due to power consumption or budget limitation.

Alternatively, detecting 3D objects from images only [16, 22, 19, 17] is a cheaper choice, but with a sacrifice of losing geometry information. Some methods guide the model to predict scene geometry using ground truth geometry, *i.e.* ground truth surface voxels [19] or TSDF [17], as supervision. However, obtaining ground truth scene geometry is troublesome. In contrast, ImVoxelNet [16] builds a 3D feature volume in the world coordinate, where each voxel center aggregate the corresponding features of its projected 2D pixels. Subsequently, 3D U-Net is applied to refine the volume features and predict bounding boxes from each voxel center. However, voxels may wrongly aggregate irrelevant image features since depth is not known. Recently, NeRF-Det [22] uses NeRF to learn a density field and predict an opacity score per voxel center to downweigh the influence of voxels in the empty space to the feature volume. However, the implicit modeling of geometry in NeRF leads to unsatisfactory performance. Moreover, the reliance of NeRF during training side-tracked the effort in making NeRF generalizable. Instead of implicitly modeling geometry with NeRF, we utilize plane-swept cost volumes for geometry-aware scene reasoning. We propose a probabilistic sampling and soft weighting mechanism to accurately decide the placement of pixel features without sampling many depth planes.

**Multi-View Depth Estimation.**    Multi-view depth estimation has long been studied in the multi-view stereo [24, 20, 25, 5, 26, 11, 7]. MVSNet [24] constructs a 3D cost volume, regularize it with a 3D CNN and regresses the depth map from the probability volume. However, MVSNet consumes large memory due the expensive 3D cost volume. Follow-up works reduce computation by replacing 3D CNN with recurrent network [23, 25] or using coarse to fine depth estimation [5, 20, 2, 7]. Recurrent methods only reduces cost regularization module to a 2D network, but still faces a large 3D cost volume to predict accurate depth. Although coarse to fine pipeline can reduce the number of sampled depth planes, it still requires to sample a certain amount of initial depth locations, *e.g.* 32 to 64 planes [7, 20], to get a reasonable coarse depth map, which still leads to intractable computation in our multi-view object detection task. Bae *et al.* [2] propose to sample extremely few number (*i.e.* 5) of initial depth planes based on a pre-trained single view depth probability distribution. It has been further applied to outdoor multi-view object detection [10]. However, both of them need the guidance of ground truth depth to learn a correct monocular depth estimation, and would otherwise fail as shown in the experiment section. Instead, we propose a probabilistic sampling and soft weighting module to bypass the large 3D cost volume to decide the 3D locations of every pixel. We also novelly utilize Gaussian Splatting to enhance our depth prediction with little computation overhead.

**3D Gaussian Splatting.**    3D Gaussian Splatting (3DGS) [9] is a recent technique for novel view rendering. It models a 3D scene explicitly with Gaussian primitives, each of which is defined by a 3D Gaussian center, covairance matrix, opacity and color. Compared to volume rendering based NeRF [12], it achieves real-time rendering with much lighter computation. To avoid per-scene optimization of 3DGS, several works [28, 3, 6, 18] propose to model a scene with pixel-aligned Gaussians Splatting (PAGS). A Gaussian primitive is predicted per pixel, and all predicted Gaussians are then combined for novel view synthesis. A key to PAGS is the accurate 3D Gaussian center which is determined by the depth estimation. Thus, instead of targeting novel view synthesis, we novelly utilize it for 3D object detection through improving depth prediction. Our method is also significantly different from NeRF-Det. While NeRF plays a major role in NeRF-Det by predicting opacity scores, we use Gaussian Splatting as a regularizer to our plane sweep algorithm with little computation overhead.

## 3   Our Method

**Problem Definition.**    The goal of multi-view indoor 3D object detection is to predict bounding box $\{\mathcal{B}\} \subseteq \mathbb{R}^7$ of objects in the 3D scene and their corresponding classes from $N$ posed images $\{I_1, \cdots, I_N\} \in \mathbb{R}^{H \times W \times 3}$. Each bounding box $\mathcal{B}$ is parameterized as $(x, y, z, w, h, l, \phi)$, where $(x, y, z)$ are the coordinates of the box center, $(w, h, l)$ are the width, height, and length, and $\phi$ is the

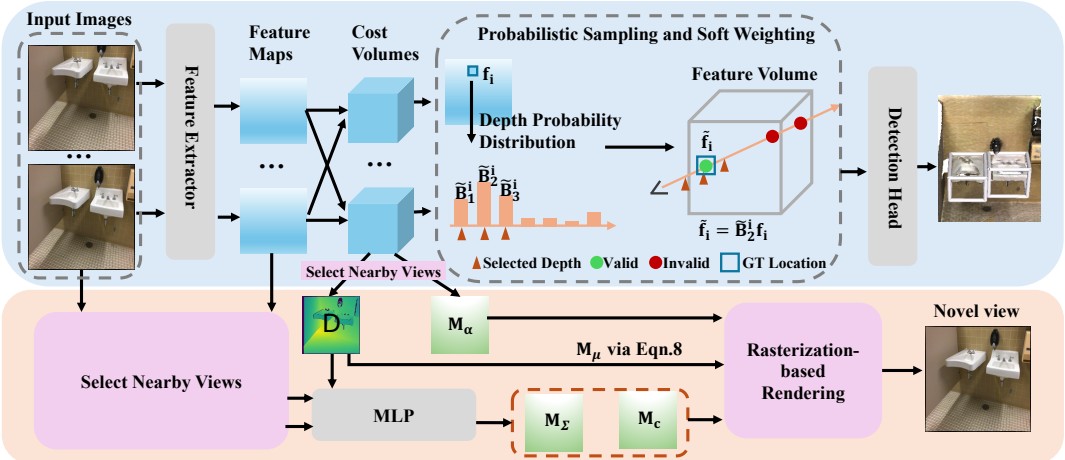

Figure 2: Overview of our MVSDet. The upper branch shows the detection pipeline with our proposed probabilistic sampling and soft weighting. The backprojected ray intersects at 3 points (shown as dots), but only the green point receives the pixel feature based on the selected depth proposals. The red points are denoted as invalid backprojection location and thus the pixel feature is not assigned to them. "GT Location" is the ground truth 3D location of the pixel. The lower branch shows the pixel-aligned Gaussian Splatting (PAGS). We select nearby views for the novel image from the images input to the detection branch and predict Gaussian maps on them. Note that PAGS is removed during testing.

rotation angle around z-axis. The intrinsic and extrinsic matrix for each input image are denoted as $K \in \mathbb{R}^{3 \times 3}$ and $P = [R \mid t] \in \mathbb{R}^{3 \times 4}$, respectively.

**Overview.** Fig. 2 shows our proposed MVSDet, a geometry-aware approach for indoor 3D object detection from posed multi-view images. Our MVSDet is built on 3D volume-based object detection (*cf.* Sec. 3.1). Instead of naively assigning the same 2D feature redundantly to every voxel center that intersect the backprojected ray, we place the pixel features according to the estimated depth from our proposed efficient plane sweep algorithm. We alleviate the costly sampling of many depth planes for accurate depth prediction by proposing a probabilistic sampling and soft weighting mechanism (*cf.* Sec. 3.2). We further utilize pixel-aligned Gaussian Splatting to enhance our depth prediction module with little extra computation cost (*cf.* Sec. 3.3). Intuitively, our depth-aware framework leads to more precise assignments of multi-view image features in the 3D volume and consequently better 3D object detection results.

## 3.1 Background: 3D Volume-Based Object Detection

Existing methods [16, 22] estimate bounding boxes from a 3D feature volume aggregated from the multi-view image features. Image features $F \in \mathbb{R}^{\frac{H}{4} \times \frac{W}{4} \times C}$ are first extracted from every input image $I \in \mathbb{R}^{H \times W \times 3}$. A 3D volume is defined in the world space with the size of $N_x \times N_y \times N_z$ voxels. Voxel center $p = [x, y, z]^{\top} \in \mathbb{R}^3$ is projected to $i$-th input image to obtain the 2D coordinate $[u_i, v_i]^{\top} \in \mathbb{R}^2$ as:

$$[u_i', v_i', d_i]^{\top} = K_i' P_i [p^{\top}, 1]^{\top}, \quad [u_i, v_i]^{\top} = [u_i'/d_i, v_i'/d_i]^{\top}, \tag{1}$$

where $K_i'$ is the scaled intrinsic matrix according the image downsampling ratio. Subsequently, 2D feature $f_i \in \mathbb{R}^C$ is assigned to p via nearest neighbour interpolation as follow:

$$f_i = \text{interpolate}\left((u_i, v_i), F_i\right). \tag{2}$$

For p that are projected outside the boundary of feature map or behind the image plane, the projection is considered as invalid and we set $f_i = 0$. Finally, it averages all valid backprojected features as the voxel feature $v = \sum_{i=1}^{n} f_i / n \in \mathbb{R}^C$, where $n$ denotes the number of valid projection.

The feature volume is refined by a 3D U-Net before being fed into the detection head to predict category, a bounding box and centerness score on each voxel location. The training losses for detection consists of focal loss for classification $\mathcal{L}_{\text{cls}}$, cross-entropy loss for centerness $\mathcal{L}_{\text{center}}$, and IoU loss for location $\mathcal{L}_{\text{loc}}$:

$$\mathcal{L}_{\text{det}} = \mathcal{L}_{\text{cls}} + \mathcal{L}_{\text{center}} + \mathcal{L}_{\text{loc}}. \qquad (3)$$

**Observation.** We observe that the feature aggregation method in existing multi-view indoor 3D object detection works causes 2D pixel features to be duplicated on voxels intersecting with the ray emitted from camera origin though the pixel as illustrated in Fig. 3. This is a result from lack of depth-awareness where voxels can end up aggregating irrelevant pixel features that are either not on the surface or occluded.

**Proposition.** To circumvent this problem, we propose: 1) an efficient plane sweep to instill depth awareness. Our efficient plane sweep consists of a probabilistic sampling and soft weighting mechanism based on a cost volume representation to evaluate the placement of pixel features on the intersected voxel centers; 2) a depth prediction regularizor based on 3D Gaussian Splatting to improve depth accuracy.

## 3.2 Efficient Plane Sweep

**Cost Volume Construction.** We build $N$ cost volumes for $N$ input images to predict $N$ depth probability distribution maps. To construct the cost volume for the $i$-th view, we set the image $\text{I}_i$ as the reference view and select 2 nearby views as the source views $\text{I}_{j \in 2\text{NB}(i)}$. A raw matching volume for $\text{I}_i$ is constructed by backprojecting source feature map $\text{F}_{j \in 2\text{NB}(i)} \in \mathbb{R}^{\frac{H}{4} \times \frac{W}{4} \times C}$ into the coordinate system defined by $\text{I}_i$ at a stack of fronto-parallel virtual planes. The virtual planes $\{\text{d}_1, \ldots, \text{d}_M\}$ are uniformly sampled on a pre-defined depth range. The coordinate mapping from the source feature map $\text{F}_j$ to the reference feature map $\text{F}_i$ at depth $\text{d}_m$ is determined by a planar homography transformation:

$$\text{q}_{j,m} = \text{K}_j \left[ \text{R}_{ij} \mid \text{t}_{ij} \right] \left[ \text{d}_m \left( \text{K}_i^{-1} \text{q} \right)^{\top}, 1 \right]^{\top}. \qquad (4)$$

where q is the homogeneous coordinate of a pixel on $\text{F}_i$, $\text{q}_{j,m}$ is the projected homogeneous coordinate of q on $\text{F}_j$. $[\text{R}_{ij} \mid \text{t}_{ij}]$ are the rotation and the translation of $\text{I}_j$ relative to $\text{I}_i$. We use Eqn. 4 to warp every source feature map $\text{F}_{j \in 2\text{NB}(i)}$ into all the depth planes and use a variance based metric [24] to generate a cost volume $\text{U} \in \mathbb{R}^{\frac{H}{4} \times \frac{W}{4} \times C \times M}$. Subsequently, we use a shallow 3D CNN to refine U to generate a probability volume (after softmax) $\text{B} \in \mathbb{R}^{\frac{H}{4} \times \frac{W}{4} \times M}$. We also predict an offset per depth bin to account for the discretization error.

Figure 3: Comparison of different feature back-projection methods. The pixel ray intersects at 4 voxel centers with the blue box denoting the ground truth 3D location of the pixel. Our method computes the placement of the pixel features based on the depth probability distribution (purple) and thus able to suppress incorrect intersections.

**Probabilistic Sampling and Soft Weighting.** Although we can place pixel features to the 3D volume by predicting depth using the weighted average based on B, it requires sampling sufficient depth planes, which is prohibitive for our task. To this end, we propose a probabilistic sampling and soft weighting mechanism to decide the placement of pixel features on its backprojected ray without the need for many depth planes.

For every pixel, we sample $k$ depth locations that score top in the corresponding distribution in B as its depth proposals. We denote their depth values as $\{\text{d}_{\text{idx}_1}, \ldots, \text{d}_{\text{idx}_k}\}$, and also use their corresponding probability score to evaluate its confidence with respect to the ground truth location. The scores are denoted as $\{\tilde{\text{B}}_{\text{idx}_1}, \ldots, \tilde{\text{B}}_{\text{idx}_k}\}$, where $\tilde{\text{B}}_{\text{idx}_k} = \text{B}_{\text{idx}_k} / \sum_{i=1}^{i=k} \text{B}_{\text{idx}_i}$. Intuitively, this is equivalent to split one depth prediction to multiple depth prediction based on the scores. As a result, it is more likely to cover the ground truth location when depth bins are insufficient.

Suppose a voxel center $\text{p} \in \mathbb{R}^3$ intersects the backprojected ray of a pixel from the $i$-th image, and the depth of p under the $i$-th camera frustum is $\text{d}(\text{p})$. We consider the projection of point p to the $i$-th

image as valid and set the indicator $g_i = 1$ when d(p) resides near any of the top-$k$ depth proposals $\{d_{\text{idx}_1}, \ldots, d_{\text{idx}_k}\}$. We assign the normalized probability score of the nearest depth proposal to p as its confidence. The corresponding pixel feature is assigned to p weighted by the confidence score. The projection is invalid when d(p) is not close to any of the depth proposals, and we set the backprojected feature from $i$-th image as 0. Thus, the feature $\tilde{f}_i \in \mathbb{R}^C$ backprojected from the $i$-th image to point p and its corresponding indicator $g_i$ are given as follows:

$$\tilde{f}_i = \begin{cases} \tilde{B}^i_{\phi(p)} f_i & \text{if d(p)} \subset \{d_{\text{idx}_1}, \ldots, d_{\text{idx}_k}\} \\ 0 & \text{otherwise} \end{cases}, \qquad g_i = \begin{cases} 1 & \text{if d(p)} \subset \{d_{\text{idx}_1}, \ldots, d_{\text{idx}_k}\} \\ 0 & \text{otherwise} \end{cases}, \quad (5)$$

where $\phi(p)$ is the index of depth proposal that is close to d(p), and $\tilde{B}^i_{\phi(p)}$ is the score of $d_{\phi(p)}$. After looping through every input image, we average all valid backprojected features for point p as its voxel feature $\hat{v} \in \mathbb{R}^C$:

$$\hat{v} = \eta_v^{-1} \sum_{i=1}^{i=N} g_i \tilde{f}_i, \quad \text{where } \eta_v = \sum_{i=1}^{i=N} g_i \tilde{B}^i_{\phi(p)}. \tag{6}$$

We also utilize the multi-view consistency to compute a surface score s for point p by averaging confidence scores $\tilde{B}^i_{\phi(p)}$ from every valid projection. We set s $= 0$ when the point does not have any valid projection, which means point p is in the free space. s is multiplied with voxel feature $\hat{v}$ as the final feature v $\in \mathbb{R}^C$ for point p as follows:

$$s = \eta_s^{-1} \sum_{i=1}^{i=N} g_i \tilde{B}^i_{\phi(p)}, \quad \text{where } \eta_s = \sum_{i=1}^{i=N} g_i, \quad v = s\hat{v}. \tag{7}$$

The adoption of s shares similar spirit with existing methods [22, 19] to decrease the influence of empty space voxels in the feature volume. Yet, we learn it directly from multi-view images instead of relying on NeRF [22] or ground truth supervision [19].

## 3.3 Enhancing Depth Prediction with Gaussian Splatting

We further utilize the recent pixel-aligned Gaussian Splatting (PAGS) [3, 28] to enhance our depth prediction module. PAGS takes in sparse views and predict a Gaussian primitive per pixel. The parameters for each primitive are center $\mu$ , Gaussian opacity $\alpha$ , covariance $\Sigma$ and color c. All the Gaussian primitives are combined together to render a novel view via rasterization-based splatting.

We select 3 nearby views per novel view from the images input to the detection branch , and predict Gaussian maps $\{M_\mu, M_\alpha, M_\Sigma, M_c\}$ for the selected views. The Gaussian center map $M_\mu \in \mathbb{R}^{\frac{H}{4} \times \frac{W}{4} \times 3}$ are directly estimated from the predicted depth based on the probability volume B as follows:

$$M_\mu(r) = o(r) + \hat{D}(r) h(r), \quad D = BG \tag{8}$$

where $G = [d_1, \ldots, d_M]^\top$ is the virtual depth planes and $D \in \mathbb{R}^{\frac{H}{4} \times \frac{W}{4} \times 1}$ is the estimated depth map. $o(r)$ is the camera origin, $\hat{D}(r)$ is the projected ray depth obtained from the depth map D, and $h(r)$ is the ray direction for pixel $r$. The Gaussian opacity map $M_\alpha \in \mathbb{R}^{\frac{H}{4} \times \frac{W}{4} \times 1}$ is predicted by taking the max probability score of B as follow:

$$M_\alpha = \max(B, dim = -1). \tag{9}$$

The Gaussian covariance map $M_\Sigma \in \mathbb{R}^{\frac{H}{4} \times \frac{W}{4} \times 16}$ and color map $M_c \in \mathbb{R}^{\frac{H}{4} \times \frac{W}{4} \times 3}$ are predicted from a MLP as follows:

$$M_\Sigma, M_c = \text{MLP}(F \parallel D \parallel \hat{I}), \tag{10}$$

where $\parallel$ denotes concatenation and $\hat{I} \in \mathbb{R}^{\frac{H}{4} \times \frac{W}{4} \times 3}$ denotes the resized image map. Following 3DGS [9], $\Sigma$ is predicted by a rotation quaternion and scaling factors. Color is predicted by spherical harmonics coefficients.

We render the image color $\hat{C}_{\text{color}}$ via alpha-blending and the rendering loss is a L2 loss as follows:

$$\mathcal{L}_{\text{render}} = ||\hat{C}_{\text{color}} - C_{\text{color}}||^2. \tag{11}$$

One of the key factors for good rendering is accurate $M_\mu$, which is directly related to correct depth estimation from our model (*cf.* Eqn. 8). The rendering loss thus iteratively guides the Gaussians to the correct 3D locations, and consequently benefits our detection pipeline. Our final loss is $\mathcal{L} = \mathcal{L}_{\text{det}} + \mathcal{L}_{\text{render}}$.

Table 1: Results on ScanNet. "GT Geo" denotes whether ground truth geometry is used as supervision during training.

| Method | GT Geo | mAP@.25 | mAP@.5 |
|---|---|---|---|
| ImGeoNet[19] | ✓ | 54.8 | 28.4 |
| CN-RMA [17] | ✓ | 58.6 | 36.8 |
| ImVoxelNet [16] | – | 46.7 | 23.4 |
| NeRF-Det [22] | – | 53.5 | 27.4 |
| Ours | – | **56.2** | **31.3** |

Table 2: Results on ARKitScenes. "GT Geo" denotes whether ground truth geometry is used as supervision during training.

| Method | GT Geo | mAP@.25 | mAP@.5 |
|---|---|---|---|
| ImGeoNet[19] | ✓ | 60.2 | 43.4 |
| CN-RMA [17] | ✓ | 67.6 | 56.5 |
| ImVoxelNet [16] | – | 27.3 | 4.3 |
| NeRF-Det [22] | – | 39.5 | 21.9 |
| Ours | – | **42.9** | **27.0** |

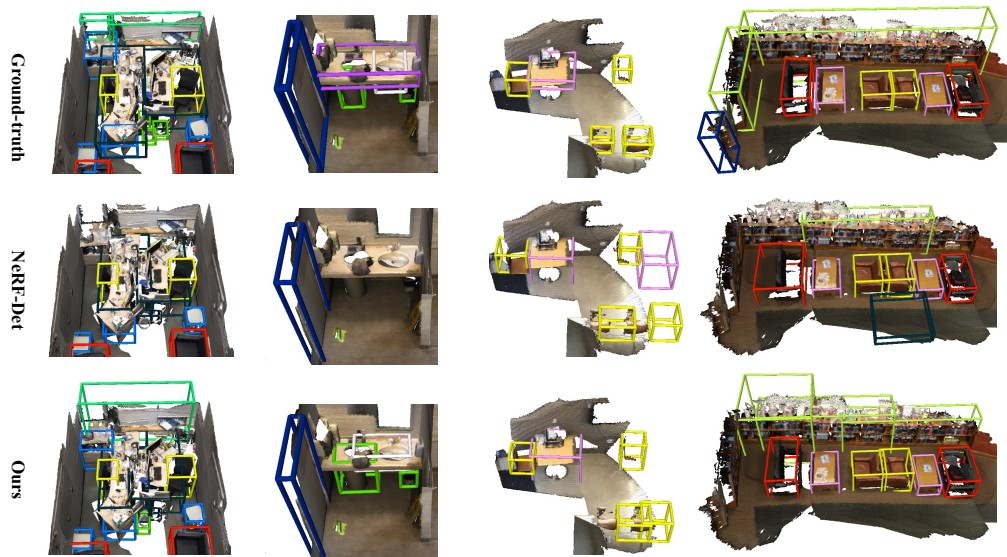

Figure 4: Qualitative comparison on ScanNet dataset. Note that the mesh is not the input to the model and is only for visualization purpose.

## 4 Experiments

### 4.1 Datasets

We conduct experiments on the ScanNet and ARKitScenes datasets. ScanNet has 1,201 and 312 scans for training and testing, respectively. We detect axis-aligned bounding boxes for 18 classes. ARKitScenes has 4,498 and 549 scans for training and testing, respectively. We detect oriented bounding boxes for 17 class. We adopt mean average precision (mAP) with thresholds of 0.25 and 0.5 as the evaluation metrics.

### 4.2 Implementation Details

We use the same feature extractor, training and testing configurations as [22] for detection. Specifically, images are resized into $(240, 320)$. During training, we input 40 images to the detection branch. During testing, the detection branch takes in 100 images and the rendering branch is removed. The size of the 3D volume is $(N_x = 40, N_y = 40, N_z = 16)$, with each voxel represents a cube of 0.16m $\times 0.16$m $\times$ 0.2m The depth range is empirically set as $[0.2m, 5m]$. The number of depth planes $M$ is set to 12 and $k = 3$ depth proposals are selected in the probabilistic sampling. We consider $d(p) \subset \{d_{idx_1}, \ldots, d_{idx_k}\}$ if $d(p)$ is within $\pm 0.2$m of any depth proposals. For the rendering branch, we select another two images as the target views that do not overlap with the images in the detection branch. We use AdamW optimizer with learning rate 0.0002, total epochs of 12 and batchsize of 1. All experiments are conducted on two NVIDIA A6000 GPUs.

Table 3: Ablation study of probabilistic sampling and soft weighting. All methods are conducted without using rendering loss.

| Probabilistic Sampling | Soft Weighting | mAP@.25 | mAP@.5 |
|:---:|:---:|:---:|:---:|
| ✓ | – | 50.0 | 24.8 |
| – | ✓ | 36.9 | 13.5 |
| ✓ | ✓ | **56.0** | **29.7** |

Table 4: Ablation study of Gaussian Splatting. $M$ denotes the number of depth planes in the plane sweep. "Gaussian" denotes using pixel-aligned Gaussian splatting. "RMSE" is the depth evaluation metric. "Memory $\Delta$" denotes the increased memory consumption during training.

| $M$ | Gaussian | mAP@.25 | mAP@.5 | RMSE | Memory $\Delta$ (GB) |
|:---:|:---:|:---:|:---:|:---:|:---:|
| 12 | – | 56.0 | 29.7 | 0.674 | **0** |
| 12 | ✓ | **56.2** | **31.3** | **0.374** | +2.6 |
| 16 | – | 55.8 | 31.1 | 0.480 | +9.4 |

### 4.3 Comparison with Baselines

We compare our method with ImGeoNet [19], CN-RMA [17], ImVoxelNet [16] and NeRF-Det [22]. We directly report their results from the CN-RMA [17] paper. Note that ImGeoNet and CN-RMA use ground truth 3D geometry as supervision during training. Tab. 1 and Tab. 2 show the results on ScanNet and ARKitScenes, respectively. It is expected that using ground truth geometry as supervision can achieve good performance. However, ground truth geometry may not be accessible and therefore we seek for a self-supervised approach that do not rely its supervision. ImVoxelNet and NeRF-Det are the two existing methods that leverage self-supervision to learn geometry for multi-view 3D detection. Compared to these works, our method achieve much better performance. It clearly shows the superiority of our efficient plane sweep method over the vanilla feature backprojection in ImVoxeNet and the density field in NeRF-Det. Fig. 1 shows the qualitative results on ScanNet. The first two columns show that our model can detect more target objects. We also find that NeRF-Det tends to detect objects in the free space (last two colums), which is due to inaccurate feature projection. In contrast, our model is able to place bounding boxes at more accurate locations.

### 4.4 Ablation Study

**Effectiveness of Probabilistic Sampling and Soft Weighting.** Tab. 3 shows the ablation stdudy of the probabilistic sampling and soft weghting (PSSW). All methods are conducted without the rendering loss $\mathcal{L}_{\mathrm{render}}$ to test the performance of PSSW alone. Removing "Probabilistic Sampling" means replacing top-$k$ sampling with a single depth estimation computed by the weighted average of depth probability volume B. We use the max probability score as the weight in this case. Removing "Soft Weighting" means replacing $\tilde{\mathrm{B}}^{i}_{\phi(\mathrm{p})}$ with value 1. As shown in the table, removing either "Probabilistic Sampling" or "Soft Weighting" causes drastic drop in the performance. Particularly, removing probabilistic sampling depth proposals drops by 19.1 at mAP@.25. It suggests that estimating accurate depth is very hard under insufficient depth planes. Furthermore, soft weighting is very important to our model as it can decrease the influence of wrong depth proposals introduced by the sampling. Overall, both probabilistic sampling and soft weighting are crucial to our model.

**Effectiveness of Gaussian Splatting for Depth Prediction.** Tab. 4 shows the ablation study for using pixel-aligned Gaussian splatting (PAGS) for detection. "RMSE" evaluates the average quality of depth prediction for the images in the detection branch. Increasing depth planes to $M = 16$ or using Gaussian Splatting both improve the depth estimation and bring similar improvement to the detection performance. However, using PAGS consumes much less memory during training than $M = 16$, *i.e.* adding only 28% memories compared to increasing depth planes. Moreover, PAGS does not bring any additional memory cost during testing because it is removed for detection. We visualize the depth maps predicted by the probability volume B in Fig. 5. The depth maps are of 1/16 size of the original image since we estimate depth on the feature map level. It shows that PAGS significantly improves the depth quality.

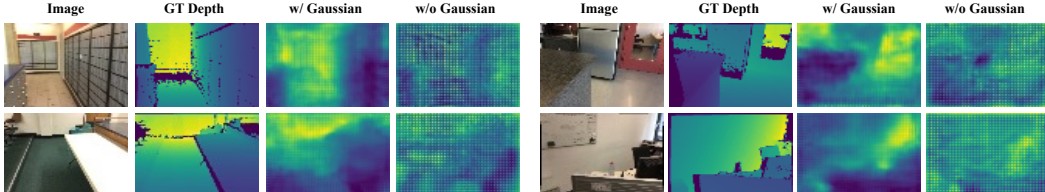

| Image | GT Depth | w/ Gaussian | w/o Gaussian | Image | GT Depth | w/ Gaussian | w/o Gaussian |

Figure 5: Depth map visualization. "GT Depth" denotes ground truth depth map. Both "w/ Gaussian" and "w/o Gaussian" use $M = 12$ depth planes.

**Analysis of the number of Top-$k$ depth proposals.** Tab. 5 shows the ablation study of number of depth proposals. Sampling top-1 depth proposal leads to severe performance decrease as the correct depth location is harder to be selected. Sampling too many depth proposals ($k = 5$) also leads to some decrease since more inaccurate locations are sampled. We thus choose $k = 3$ in our model.

Table 5: Ablation study of Top-$k$ depth proposals.

| k | mAP@.25 | mAP@.5 |
|---|---------|--------|
| 1 | 49.3 | 24.4 |
| 3 | **56.2** | **31.3** |
| 5 | 55.5 | 29.8 |

**Comparison with Depth Estimation Methods.** Fig. 6 shows the comparison of different depth prediction methods to the detection performance. MVSNet [24] performs one-time plane sweep by using $M = 16$ depth planes. BEVStereo [10] performs iterative depth estimation by sampling depth planes according to a monocular depth estimation. We set the number of iterations to 3 and the number of depth planes in each iteration to 5. We remove ground truth depth supervision for BEVStereo for fair comparison. "Ground-truth Depth" denotes placing pixel feature on the 3D volume according to the ground-truth depth location, which is the upper bound of our method. We show the results of our approach using different number of depth planes ($M = 12$ and $M = 8$). MVSNet performs badly even though it samples

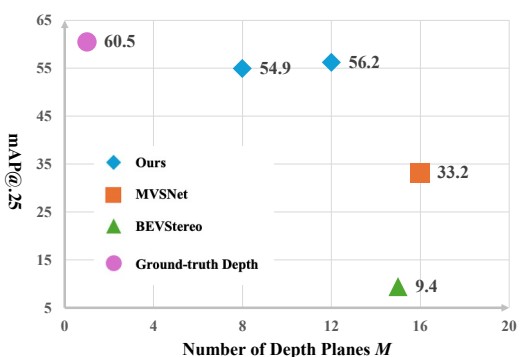

Figure 6: Comparison of different depth prediction methods on 3D object detection on ScanNet.

more planes than us. This is because MVSNet requires sufficient depth planes to estimate depth correctly. BEVStereo also fails because it requires the ground truth depth supervision to learn a good initial monocular depth estimation. In contrast, our model achieve close performance as the "Ground-truth Depth" using only 12 or even 8 depth planes. This demonstrates that our approach efficiently learns geometry without sampling many depth planes.

**Time and memory comparison.** Tab. 6 shows the comparison of time and memory in the training and testing stages on ScanNet, respectively. We omit comparison with ImGeoNet since it does not release any code. All models are ran on $2 \times$ A6000 GPUs. Due to the complexity of CN-RMA (as mentioned in Sec 3.5 of their paper), it requires much longer time to train and evaluate than other models. Furthermore, CN-RMA consumes much more memory in the training stage because it requires joint end-to-end training of the 3D reconstruction and detection network. Although NeRF-Det is efficient in time and memory of the training and testing stages, their performance is much worse than ours as shown in Tab. 1 and 2.

Table 6: Time and memory comparison in training and testing stages on ScanNet dataset, respectively.

| Method | Train | | Test | |
|--------|-------|-----------|------|-----------|
| | Time(hrs) | Memory(GB) | Time(min) | Memory(GB) |
| CN-RMA[17] | 121 | 43 | 10 | 12 |
| NeRF-Det[22] | **7** | **13** | **2** | **12** |
| Ours | 18 | 35 | 3 | 28 |

# 5 Conclusion

In this paper, we propose MVSDet for multi-view image based indoor 3D object detection. We design a probabilistic sampling and soft weighting mechanism to decide the placement of pixel features on the 3D volume without the need of the computationally sampling of many depth planes. We further introduce the use of pixel-aligned Gaussian Splatting to improve depth prediction with little computation overhead. Extensive experiments on two benchmark datasets demonstrate the superiority of our method.

# 6 Limitation

Similar to existing multi-view stereo methods [24, 20], feature matching would fail on texture-less or reflective surfaces. One possible solution is to combine with monocular depth estimation [2]. However, estimating monocular depth is non-trivial and we leave it for future research.

**Acknowledgement.** This research work is supported by the Agency for Science, Technology and Research (A*STAR) under its MTC Programmatic Funds (Grant No. M23L7b0021).

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

# A    Appendix / Supplemental Material

## A.1    Qualitative Results on ARKitScenes Dataset

Fig. 2 shows the qualitative results on ARKitScenes dataset. Compared to NeRF-Det [22], we are able to detect more target objects.

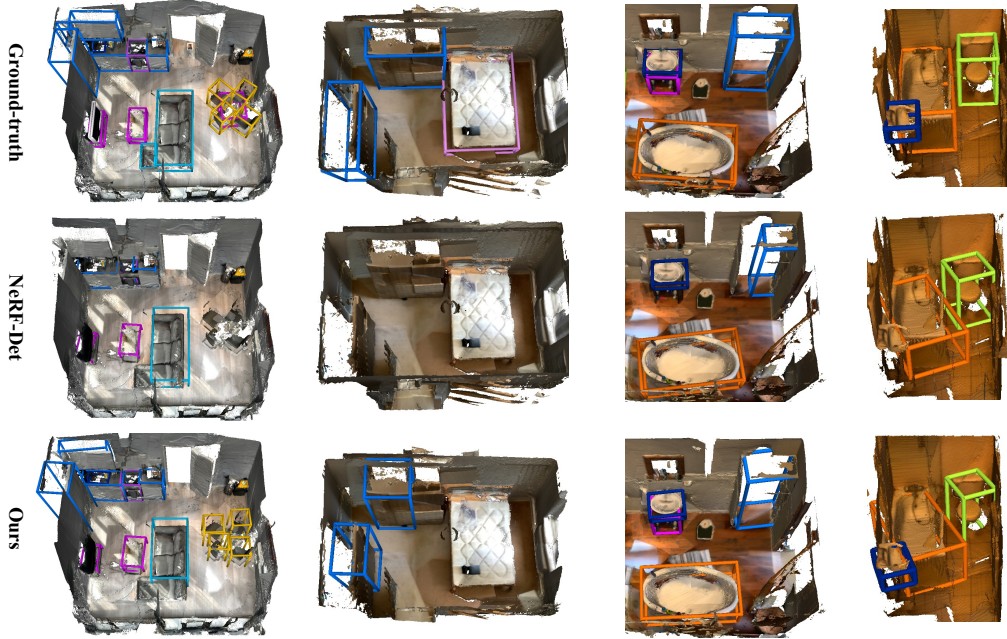

Figure 7: Qualitative comparison on ARKitScenes dataset. Note that the mesh is not the input to the model and is only for visualization purpose.

## A.2    Novel View Synthesis Results

Fig. 8 shows the novel view synthesis results on ScanNet test dataset. Our model gives reasonable rendering results, indicating that the Gaussian Splatting module successfully learn the geometry. Note that the Gaussian Splatting module is only a regularizer to our plane sweep algorithm instead of the determining factor to learn geometry like NeRF in NeRF-Det [22].

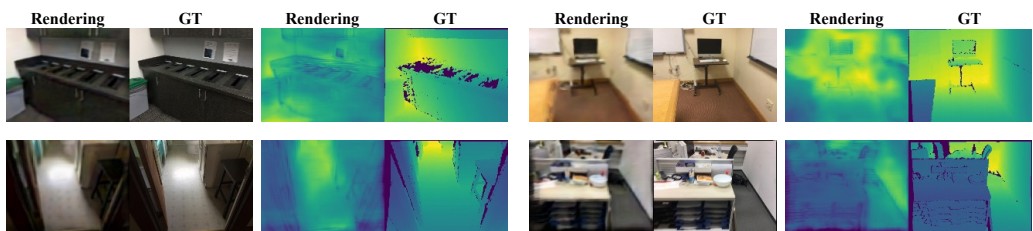

Figure 8: Novel view synthesis results on ScanNet dataset."Rendering" denotes the rendered image / depth from our Gaussian Splatting module. "GT" denotes the ground-truth image /depth of the novel view.

Table 7: Per-class results under AP@0.25 on ScanNet dataset.

| Method | cab | bed | chair | sofa | tabl | door | wind | bkshf | pic | cntr | desk | curt | fridg | showr | toil | sink | bath | other |
|---|---|---|---|---|---|---|---|---|---|---|---|---|---|---|---|---|---|---|
| NeRF-Det | **42.3** | **84.6** | 75.9 | 78.5 | **56.3** | 33.4 | 21.4 | 49.9 | 2.4 | **50.6** | **73.9** | 21.3 | 54.3 | 62.5 | 90.9 | **57.7** | 75.5 | 32.3 |
| Ours | 40.5 | 82.4 | **79.2** | **80.2** | 55.6 | **40.3** | **25.4** | **60.9** | **3.5** | 47.3 | 73.4 | **28.9** | **64.6** | **64.1** | **94.8** | 52.1 | **76.7** | **41.8** |

Table 8: Per-class results under AP@0.5 on ScanNet dataset.

| Method | cab | bed | chair | sofa | tabl | door | wind | bkshf | pic | cntr | desk | curt | fridg | showr | toil | sink | bath | other |
|---|---|---|---|---|---|---|---|---|---|---|---|---|---|---|---|---|---|---|
| NeRF-Det | **15.8** | **73.1** | 45.3 | 40.6 | **39.5** | 8.1 | 2.0 | 20.3 | 0.2 | **13.8** | 42.5 | 5.3 | 25.3 | 10.0 | 63.0 | 26.0 | 49.1 | 12.7 |
| Ours | 14.9 | 71.4 | **48.9** | **54.4** | 38.8 | **9.5** | **3.1** | **29.6** | **0.8** | 9.8 | **48.5** | **5.6** | **40.2** | **10.2** | **77.3** | **29.0** | **52.9** | **17.7** |

## A.3   Per-Class Performance

Tab. 7, Tab. 8, Tab. 9 and Tab. 10 are the per-class results under AP@0.25 and AP@0.5 on ScanNet and ARKitScenes datasets, respectively.

Table 9: Per-class results under AP@0.25 on ARKitScenes dataset.

| Method | cab | fridg | shlf | stove | bed | sink | wshr | tolt | bthtb | oven | dshwshr | frplce | stool | chr | tbl | TV | sofa |
|---|---|---|---|---|---|---|---|---|---|---|---|---|---|---|---|---|---|
| NeRF-Det | 34.7 | 61.1 | 30.7 | 9.4 | 73.2 | 29.9 | **62.6** | 77.2 | 86.4 | 45.0 | 7.4 | 2.1 | 12.1 | 46.4 | 38.3 | 0.1 | 55.5 |
| Ours | **42.7** | **65.6** | **34.6** | **12.1** | **77.9** | **35.5** | 61.5 | **78.9** | **86.9** | **51.5** | **13.6** | **5.4** | **13.2** | **50.0** | **40.7** | **0.2** | **59.0** |

Table 10: Per-class results under AP@0.5 on ARKitScenes dataset.

| Method | cab | fridg | shlf | stove | bed | sink | wshr | tolt | bthtb | oven | dshwshr | frplce | stool | chr | tbl | TV | sofa |
|---|---|---|---|---|---|---|---|---|---|---|---|---|---|---|---|---|---|
| NeRF-Det | 10.8 | 48.0 | 5.7 | 0.6 | 36.1 | 7.9 | 46.3 | 60.8 | 64.9 | 21.0 | 5.6 | 0.0 | 2.9 | 18.8 | 14.1 | 0.0 | 28.2 |
| Ours | **17.5** | **50.6** | **9.2** | **1.9** | **51.9** | **9.9** | **51.6** | **65.0** | **70.6** | **27.8** | **9.3** | **1.8** | **6.6** | **29.1** | **20.2** | 0.0 | **36.5** |

