# OpenReview forum: "MVSDet: Multi-View Indoor 3D Object Detection via Efficient Plane Sweeps"
_NeurIPS.cc/2024/Conference — NeurIPS 2024 poster_

### Official Review · Reviewer_ZuP3 · 2024-07-02

**Soundness:** 3
**Presentation:** 3
**Contribution:** 3
**Rating:** 6
**Confidence:** 4

**Summary:**

This paper mainly focuses on the problem of 3D object detection form multi-view images. It introduces MVSNet-like method for depth prediction, brings out probabilistic sampling, soft weighting and pixel-aligned Gaussian Splatting to improve the correctness, robustness of depth prediction, especially with sparse images. Therefore, it improves the performance for 2D features to be projected to the 3D space, therefore improves the performance of 3D object detection.

**Strengths:**

A substantive assessment of the strengths of the paper, touching on each of the following dimensions: originality, quality, clarity, and significance. We encourage reviewers to be broad in their definitions of originality and significance. For example, originality may arise from a new definition or problem formulation, creative combinations of existing ideas, application to a new domain, or removing limitations from prior results. You can incorporate Markdown and Latex into your review. See https://openreview.net/faq.

This paper is clear-written. It introduces probabilistic sampling, soft weighting and pixel-aligned Gaussian Splatting to improve the performance of MVSNet-based depth prediction to improve the performance of 3D object detection without the need of ground truth geometry.

**Weaknesses:**

1. The 2D feature extractor, the detection head, and the detection head of this work is not clearly described.
2. The MVSNet-like depth prediction with probabilistic sampling and soft weighting, as well as the pixel-aligned Gaussian Splatting are all common methods or from previous works, I consider the novelty to be limited.
3. Also, I do not agree with the opinion of the article that ground truth geometry for supervision on the training stage is difficult to obtain. As far as I know, To obtain supervision for 3D object detection (AABB or OBB bounding boxes), you need to use a lidar or a RGBD camera to obtain the geometry of a 3D scene, and then label the bounding boxes.

**Questions:**

1. I do not know about the 2D feature extractor, the detection head, and the detection head of this work, are they the same as NeRF-Det?
2. Please further explain the novelty of your method, or your novel discovery that leads you to your method.
3. I do not agree with the opinion of the article that ground truth geometry for supervision on the training stage is difficult to obtain, please further explain this claim, since your results are not better than CN-RMA and ImGeoNet that used ground truth geometry for supervision.
4. Following my question 3, introducing ground truth geometry may cause significant time and memory consumption in training, and makes the network far more complicated and therefore increases time and memory cost in inference. Could you give me detailed time and memory consumption of your own method and all the baselines you have run, on both training and inference stages?

**Limitations:**

Yes, the authors have addressed the limitations.

---

> ### Author Rebuttal · Authors · 2024-08-06
>
> **[R3/Q1] Feature extractor and detection head.** Yes, they are the same with NeRF-Det. We use the ResNet50 to extract image features at multiple stages, and fuse them via a feature pyramid network as the final feature map for each image. The 3D U-Net (Line 149) outputs feature maps at three scales, all of which are sent into the detection head. The detection head consists of three 3D convolutional layers for classification, location, and centerness prediction, respectively. From each voxel center and each of the three scales, the head estimates a class probability, a centerness score, and 3D bounding box offset. Please refer to NeRF-Det for more details.
>
> **[R3/Q2.1] Limited novelty.** We do not agree. **(1)** Compared to other MVS models (which predict one depth location obtained from probability weighted sum over all depth planes), we are the first to _probabilistically sample_ multiple top-scored depth locations. As agreed by **Reviewer YYFF**, our probabilistic sampling and soft weighting shows strong superiority over the normal mvs models when there is only few number of depth planes. **(2)** The pixel-aligned Gaussian Splatting is originally only used in the novel view synthesis. We are the first to apply it in the detection task to regularize the depth prediction with light computation overhead.
>
> **[R3/Q2.2] Novel discovery leading to our method.** As shown in Fig 1 of our main paper, NeRF-Det shows many wrong backprojections of 2D pixel features to the points in the free space due to the inaccurate geometry learned from NeRF. Therefore, we propose to use plane sweep to better estimate geometry. Compared to NeRF which does not have sufficient surface constraints (only predict a density score per point), the plane sweep approach can accurately predict the surface. However, the standard plane sweep method require sampling many depth planes to estimate the depth accurately, which leads to intractable computation for our multi-view 3D detection task. To tame computation complexity while maintaining accuracy, we propose probabilistic sampling and soft weighting together with the novel use of pixel-aligned Gaussian Splatting.
>
> **[R3/Q3] GT geometry is needed for labeling bboxes and thus should be used in supervision.** We disagree. 3D bounding boxes can be annotated _without_ using ground-truth (dense) geometry obtained from lidar or rgb-d cameras. An example is the Objectron Dataset **[1]**, where it annotates the 3D bounding boxes on a _sparse point cloud_ obtained via feature tracking on a AR device, and refines bounding boxes by re-projecting them onto the multi-view images (see Section 3.2 and 3.3 of their paper).
> Due to the requirement of dense geometry to compute TSDF or surface voxels for supervision, CN-RMA and ImGeoNet are largely restricted to certain datasets such as Scannet and ARKitScenes, where tedious heavy postprocessing of the raw data are needed to ensure high quality dense geometry. In contrast, our proposed method offers better versatility on datasets without dense geometry since we do not rely on the GT geometry for supervision.
> **[1] Objectron: A Large Scale Dataset of Object-Centric Videos in the Wild with Pose Annotations. CVPR 2021**
>
> **[R3/Q4] Time and memory comparison.** **Tab. R3-ZuP3/Q4** of attached PDF shows the comparison of time and memory in training and testing stages on the ScanNet, respectively. We omit comparison with ImGeoNet since it does not release any code. All models are run on 2 A6000 GPUs. Due to the complexity of CN-RMA (as mentioned in Sec 3.5 of their paper), it requires much longer time to train and evaluate than other models. Furthermore, CN-RMA consumes much more memory in the training stage because it requires joint end-to-end training of the 3D reconstruction and detection network. Although NeRF-Det is efficient in time and memory of training and testing stages, their performance is much worse than ours as shown in Table 1 and 2 of our main paper.

---

> > ### Comment · Reviewer_ZuP3 · 2024-08-08
> >
> > The rebuttal clearly addresses my questions. While its performance does not surpass CN-RMA, it is efficient in terms of time and memory cost during both training and testing stages. I strongly recommend including Tab.R3-ZuP3/Q4 in the final version. I would like to raise my rating.

---

### Official Review · Reviewer_YYFF · 2024-07-09

**Soundness:** 3
**Presentation:** 2
**Contribution:** 3
**Rating:** 6
**Confidence:** 4

**Summary:**

The manuscript proposes MVSDet, a multiview 3d object detection model that is evaluated on indoor scene datasets. Multiview information is lifted to 3D via an efficient per-frame depth sampling scheme. The most probable top-k depth values per pixel are used to lift 2D features into a global feature volume in a weighted way. Based on the thus accumulated 3d feature volume a 3d network regresses 3d bounding box parameters. In order to regularize the depth regression which is essential for constructing a occlusion-aware 3d feature volume, the manuscript proposes leveraging pixel-aligned Gaussian splats to construct a rendering loss against nearby views. Both the probabilistic depth sampling and the rendering loss are shown to contribute significantly to the performance of the model. Overall the model outperforms other models that do not use GT depth supervision during training.

**Strengths:**

The probabilistic depth estimation shows strong performance improvement over the dense planesweep depth approach when considering memory. Tab 3 is very effective in convincing me of the need for the probabilistic depth sampling. I do wonder how the numbers change without the regressed depth offset correction?

Showing that Gaussian Splat-based rendering loss supports the performance of object detection model is useful since not all 3d object detection datasets do have surface GT for training. Tab 4 shows a small improvement for the more strict mAP threshold.

These two contributions are strongly supported by the ablation studies in the experiment section and useful to be shared with the community.

**Weaknesses:**

I do wonder how much the difference is to supervising with GT depth instead of gaussian splats. (The experiment in Fig 6 is similar but not quite the same since placing features at GT depth locations does not need the probabilistic depth model to regress the depth). This last experiment might drive home the effectiveness of Gaussian splats and allow direct comparison to the ImGeoNet and CN-RMA related works in Tab 1 and 2.

The writing and illustrations are mostly clear and support the understanding of the manuscript. There are quite a few open questions (see questions) that should be addressed to improve the presentation of the method.

**Questions:**

- Fig 1: it is kind of hard to see what is going on with the red points in the renderings.
- Fig 2: "ray intersects at 3 points (shown as dots)" -- do you mean as red triangles?
- Fig 2: the arrows in the gaussian splats orange box of the diagram are not very clear. This could be polished more.
- l 151: it is unclear what is ment by 27 location canidates are selected for each target object?
- l 184: how is the depth offset predicted in more detail and how is it used during sampling? Is it used for example in Eq 8 to adjust the depth planes per pixel?
- ablation that does supervise the proposed model with gt depth instead of Gaussian splats rendering?
- Tab 4: is the memory change per batch? (I assume since B=1)?

**Limitations:**

The limitations are addressed adequately in the paper.

---

> ### Author Rebuttal · Authors · 2024-08-06
>
> **[R2/Q1] No depth offset and how to use depth offset.** The first row of **Tab. R2-YYFF/Q1** of attached PDF shows the result of removing depth offset, which is worse than our model. Please refer to **R1/Q6** on how to predict and use depth offset.
>
> **[R2/Q2] Use GT depth as supervision.** **Tab. R2-YYFF/Q2** of attached PDF shows the ablation study of replacing Gaussian splats with ground truth depth supervision. The performance of our model is very close to using ground truth depth as supervision, which strongly verifies the effectiveness of Gaussian Splats. In addition, the 'GT Depth' model still cannot directly be compared with ImGeoNet or CN-RMA because the two models require the dense 3D geometry to supervise, which need tedious 3D reconstruction procedures on top of the raw RGB-D data.
>
> **[R2/Q3] Red points in Fig 1.** The red points are the voxel centers of the 3D volume for detection (see Sec 3.1 in our main paper). They never appear in the rendering branch. The rendering is done by splatting the Gaussian primitives predicted from the selected nearby views.
>
> **[R2/Q4,Q5] Fig 2.** The red triangles denote the selected depth locations by our probabilistic sampling. The three points (one green and two red) are the intersections of the ray in the 3D volume. Only the green point receives the corresponding pixel feature because it resides near the selected depth locations. The red points are the invalid backprojection locations. We will refine Fig 2 in the final version.
>
> **[R2/Q6] 27 locations.** Following NeRF-Det and ImVoxelNet, we apply center sampling to select candidate voxels in the 3D volume that are responsible to regress each target object. For each target object, we sample 27 voxel locations that are closest to the target object center. We will add the clarification to this part in the final paper.
>
> **[R2/Q7] Memory change per batch.** Yes. We set batch size to 1 and Tab 4 of main paper shows the memory change per batch.

---

> > ### Comment · Reviewer_YYFF · 2024-08-09
> >
> > Thank you for addressing my questions and the two evaluations that support the claims in the paper (depth offset helps, GS rendering is nearly the same as GT depth supervision). From looking at the other reviews I dont see any other problems that were not addressed. The consensus seems to move to weak accept which I support.

---

### Official Review · Reviewer_N557 · 2024-07-12

**Soundness:** 3
**Presentation:** 3
**Contribution:** 2
**Rating:** 6
**Confidence:** 3

**Summary:**

This work presents a method for multi-view 3d object detection. The method computed a MVS cost volume using a few planes, it then samples k likely depth values per pixel and builds a 3d feature volume based on the voxels close to the samples depth values weighted by their confidence. Additionally, during training pixel aligned gaussian splatting (GS) is used to provide an additional rendering loss to guide the depth estimation.

**Strengths:**

The proposed 3d feature volume construction seems to outperform the existing baselines.
The use of GS during training seems to also slightly improve the 3d object detection.

**Weaknesses:**

It is unclear how the nearby views are selected. Is it the same as in the existing works?
To evaluate the contribution of the probabilistic sampling alone it would be good to add a line in table 3 without probabilistic sampling and soft weighting.
What is the benefit of using the PAGS rendering loss over the classical MVS photometric losses?

**Questions:**

Why does it not improve with more planes? Is the 3d grid too coarse?
It seems that 2 views are used for MVS, but 3 for GS. Why not use more?
How is the depth offset predicted? Like in MVSNet? Is it ever used for the object detection?

**Limitations:**

Limitations were mentioned.

---

> ### Author Rebuttal · Authors · 2024-08-07
>
> **[R1/Q1] How to select nearby views.** We compute the Euclidean distance between the camera location of reference view and input views to find the nearby views.
>
> **[R1/Q2] No probabilistic sampling or soft weighting.** By comparing Row 2 and 3 of Tab 3 in our paper, we can already evaluate the effectiveness of probabilistic sampling as it shows severe performance decrease without probabilistic sampling. We also conduct an experiment of removing probabilistic sampling and soft weighting in **Tab. R1-N557/Q2** (attached PDF). Compared with Row 2 of the same table, not using probabilistic sampling has large detection performance drop, which again verifies the effectiveness of probabilistic sampling.
>
> **[R1/Q3] Compare with photometric loss.** The first row of **Tab. R1-N557/Q3** (attached PDF) shows the result of replacing PAGS with photometric loss. Photometric loss assumes the consistent pixel colors across nearby views and would fail in the case of occlusion (e.g. cluttered objects) or non-Lambertian surface (e.g. the varnish layer of the furniture), which is common in the indoor scene. Consequently, it only brings marginal improvement. In contrast, the PAGS can reconstruct the scene with 3D Gaussians and the spherical harmonics of each Gaussian model the view-dependent colors.
>
> **[R1/Q4] Why not improve with more planes?** We conjecture the saturation of performance at mAP@.25 vs. increase of performance at mAP@.5 from 12 to 16 planes in **Tab. R1-N557/Q4** (attached PDF) is likely due to the less strict evaluation at mAP@.25, which requires less accurate bounding box localization and consequently does not gain more information from the increase in the number of depth planes.
>
> **[R1/Q5] why not use more views?** We follow MVSNet[23] to use 2 views for MVS. We set 3 views in GS because we empirically find 3 nearby views is enough to cover the local area and adding more views does not bring significant performance as shown in **Tab. R1-N557/Q5** of attached PDF. 'GS=5' means using 5 views for GS and 2 views for MVS.
>
> **[R1/Q6] How to predict and use depth offset.** The depth offset is predicted through a MLP on top of the refined cost volume for each depth bin. Each offset is added to its corresponding depth bin to adjust the discrete depth planes. They are used in the probabilistic sampling of $\text{d}_{\text{idx}_k}$ (Line 191) and the depth map D (Eqn 8) of Gaussian Splatting.

---

> > ### Comment · Reviewer_N557 · 2024-08-09
> > **Answer to Rebuttal**
> >
> > Thank you for your detailed answers and additional evaluations. As all of my questions have been answered I would consider raising my rating to weak accept, if no further discussions arise.

---

### Author Rebuttal · Authors · 2024-08-06

We thank all reviewers for their affirmation of the effectiveness of the proposed probabilistic sampling and soft weighting and the use of Gaussian Splatting for multi-view 3D object detection without using ground-truth geometry as supervision.  We strongly agree with **Reviewer YYFF** that the two contributions deserve to be shared with the community.  **All the experiment tables are included in the attached PDF**.

---

### Decision · Program_Chairs · 2024-09-25

**Decision:**

Accept (poster)

**Comment:**

All three reviewers appreciate the methods, mainly its efficiency. Issues and questioned raised by the reviewers were clarified in the rebuttal and the discussion.